# UnEBOLT: A Unified Model for EEG-to-BOLD Translation and Functional Connectivity Reconstruction

**Yamin Li**[1]                                         YAMIN.LI@VANDERBILT.EDU
[1] *Vanderbilt University*

**Ange Lou**[1]                                        ANGE.LOU@VANDERBILT.EDU
**Chang Li**[1]                                        CHANG.LI@VANDERBILT.EDU
**Shiyu Wang**[1]                                   SHIYU.WANG.1@VANDERBILT.EDU
**Haatef Pourmotabbed**[1]            HAATEF.POURMOTABBED@VANDERBILT.EDU
**Ziyuan Xu**[1]                                     ZIYUAN.XU@VANDERBILT.EDU
**Shengchao Zhang**[1,2]                SHENGCHAO_ZHANG@BROWN.EDU
[2] *Rhode Island Hospital (Brown University Health)*

**Dario J. Englot**[1,3]                         DARIO.ENGLOT@VUMC.ORG
[3] *Vanderbilt University Medical Center*

**Soheil Kolouri**[1]                            SOHEIL.KOLOURI@VANDERBILT.EDU
**Daniel Moyer**[1]                             DANIEL.MOYER@VANDERBILT.EDU
**Roza G. Bayrak**[1]                          ROZA.G.BAYRAK@VANDERBILT.EDU
**Catie Chang**[1]                              CATIE.CHANG@VANDERBILT.EDU

**Editors:** Accepted for publication at MIDL 2026

## Abstract

Functional magnetic resonance imaging (fMRI) provides high-resolution, whole-brain dynamic information, but is costly and immobile, limiting its utility in low-resource settings. EEG-to-fMRI translation via deep learning offers a promising alternative, enabling access to deep brain activity from scalp EEG signals in naturalistic settings. However, current state-of-the-art methods for EEG-to-fMRI translation require training separate models for each brain region, limiting efficiency and scalability. Here, we introduce **UnEBOLT**[1], a **Un**ified model for **E**EG-to-**BOL**D **T**ranslation. UnEBOLT is an end-to-end framework that predicts whole-brain fMRI time series from EEG by adaptive multi-region decoding within a single model. This approach enables efficient and comprehensive inference while also reconstructing subject-specific functional connectivity matrices, a representation that provides insight into neuronal interactions and which has been successfully utilized for clinical biomarkers. Our results show that UnEBOLT achieves comparable performance to dedicated ROI-specific models while scaling to multi-region prediction. Additionally, the reconstructed fMRI time series enable functional connectivity estimation, which may have broad applications in neuroscience.

**Keywords:** EEG, fMRI, EEG-to-fMRI synthesis

---

1. https://github.com/neurdylab/UnEBOLT

## 1. Introduction

Functional magnetic resonance imaging (fMRI) allows for studying whole-brain dynamics with high spatial resolution, providing insights into large-scale neural circuits that underlie cognition, perception, and behavior (Van Den Heuvel and Pol, 2010). Functional connectivity (FC), which captures temporal correlations between blood-oxygen-level-dependent (BOLD) signals across different brain regions, has been instrumental in understanding brain networks in both healthy and clinical populations, making it a valuable tool for neuroscience and medical research. However, the widespread use of fMRI is hindered by its high cost and immobility, limiting accessibility in neuroscience and medical settings.

Electroencephalography (EEG), in contrast, is a portable and cost-effective modality with high temporal resolution. However, EEG lacks the spatial specificity of fMRI, making it difficult to infer activity in deep brain structures or resolve large-scale connectivity patterns with the same precision (Cohen, 2017; Chang and Chen, 2021). To bridge this gap, EEG-to-fMRI translation via deep learning has recently emerged as a promising direction, enabling the reconstruction of fMRI signals from scalp EEG recordings. Prior works such as (Liu and Sajda, 2023a,b; Calhas and Henriques, 2022; Lanzino et al., 2024; He et al., 2025) have primarily focused on reconstructing spatial features of fMRI volumes and task activation maps. While these studies demonstrate promising whole-brain spatial reconstruction capability, they generally lack a systemic evaluation of temporal reconstruction fidelity, higher-order functional features such as functional connectivity (FC), or cross-dataset generalization. On the other hand, studies such as (Kovalev et al., 2022; Li et al., 2024a) have investigated fMRI time series reconstruction in deep brain regions using sequence-to-sequence models, evaluating performance by computing the temporal correlation with ground-truth. However, these studies used small sample sizes and relied on subject-specific training, restricting their generalizability.

Furthermore, most existing research has focused on EEG-to-fMRI synthesis during task-based and eyes-open resting-state conditions, leaving the fully eyes-closed resting state largely unexplored. Resting-state fMRI is of notable importance because it reflects the brain's spontaneous and intrinsic activity in the absence of explicit task constraints, thereby capturing more diverse neural dynamics that are more challenging to decode. Moreover, resting-state paradigms impose minimal cognitive or behavioral demands on participants and are therefore widely adopted across large-scale and clinical neuroimaging studies, including populations for whom task-based experiments may be impractical or unreliable. A recent advancement, NeuroBOLT, introduced a framework that learns multi-dimensional representations of EEG windows to predict corresponding fMRI data points, achieving state-of-the-art performance on eyes-closed resting-state data. However, NeuroBOLT, along with (Kovalev et al., 2022; Li et al., 2024a), require separate models to be trained for each brain region, which is computationally intensive for whole-brain inference. Additionally, only a small subset of brain regions was examined in these studies (Kovalev et al., 2022; Li et al., 2024a,b), leaving open the question of EEG's predictive power for fMRI signals across the whole brain. These limitations highlight the need for a more efficient, whole-brain approach capable of learning shared representations across regions while preserving fMRI's rich temporal structure.

Here, we introduce a novel and scalable EEG-to-fMRI translation framework for whole-brain inference. The key contributions of this work are:

- **Efficient end-to-end EEG-to-fMRI translation framework.** Our framework reconstructs whole-brain fMRI dynamics within a single model, significantly reducing computational overhead while maintaining signal reconstruction fidelity.

- **Adaptive Multi-Region Decoding.** Our model dynamically refines ROI-specific representations by conditioning fMRI predictions on global EEG features, ensuring both regional specificity and whole-brain coherence. A multi-objective loss function integrates temporal and spatial correlation constraints, preserving neural dynamics and functional relationships across the brain.

- **Cortex-wide evaluations across multi-region and multiple conditions.** We performed extensive experiments on both subject-specific and cross-subject learning in ROI-level and brain-network-level. Further, we assess functional connectivity (FC) reconstruction at multiple scales, making this the first deep learning model to be evaluated on FC recovery, and additionally evaluated zero-shot generalization to task-condition datasets collected under different acquisition settings.

## 2. Methods

### 2.1. Problem Definition

Four-dimensional fMRI data are often summarized using brain parcellation techniques that average voxel-wise signals within predefined regions-of-interest (ROIs). This approach offers a practical trade-off between spatial resolution and computational efficiency, yielding higher signal-to-noise ratio (SNR) and more stable time courses while retaining the organizational structure of the brain. The resulting parcellated fMRI can be expressed as $Y \in \mathbb{R}^{P \times K}$, where each row corresponds to the time series of a specific ROI, with $K$ denoting the total number of fMRI time points. EEG windows spanning a duration $T$ approximating the hemodynamic response function (HRF) before each fMRI frame are extracted as input, represented as $X \in \mathbb{R}^{C \times T}$, where $C$ is the number of electrodes. Thus, given the model $f_\theta(.)$, the fMRI prediction is formulated as $\hat{Y}_t = f_\theta(X_{t-T:t-1})$, where $\hat{Y}_t \in \mathbb{R}^P$ is the reconstructed fMRI frame at time index $t$.

### 2.2. Model Architecture

We propose an end-to-end framework, UnEBOLT, for predicting multi-region fMRI signals from EEG recordings. Our approach learns a comprehensive mapping from EEG to fMRI data while preserving the inter-regional dependencies characteristic of fMRI data. The framework is composed of three key modules: (1) **Multi-dimensional Encoder** (2) **Gated Adaptive Fusion Module** (3) **ROI-specific Representation Learning Module**: a decoder that comprises 3 key components - i. Learnable ROI Lookup Table, ii. ROI Representation Embedder, iii. ROI Prediction Head. Together, these modules translate EEG-derived features to region-aware fMRI predictions.

**Multi-dimensional Encoder.** We leverage the multi-dimensional EEG encoder (Li et al., 2024b) as backbone to extract complementary *spatiotemporal* and *multi-scale spectral*

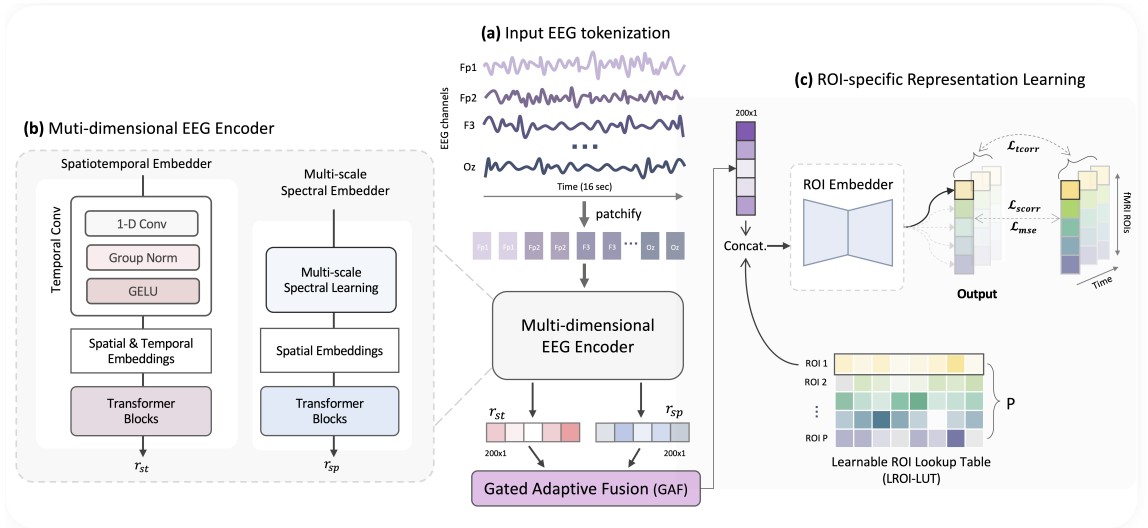

Figure 1: UnEBOLT pipeline. We tokenize EEG windows into patches and use our encoder to extract spatiotemporal and spectral embeddings, which are adaptively fused and refined by ROI-specific representation learning module for prediction.

embeddings from raw EEG signals $X \in \mathbb{R}^{C \times T}$. **Spatiotemporal Features:** We first segment $X$ using a window of length $w$ and a stride of $w$, yielding patches $x_{c,k} \in \mathbb{R}^w$ for each channel $c = 1, \ldots, C$ and patch index $k = 1, \ldots, \lfloor T/w \rfloor$. Each patch is mapped to a $d$-dimensional embedding $e_{c,k}$ through a convolutional temporal encoder and further enriched with trainable temporal ($te_k$) and spatial ($se_c$) embeddings: $e_{\text{st}}^{c,k} = e_{c,k} + te_k + se_c$. These embeddings are processed by a Transformer encoder and subsequently average-pooled to form the spatial–temporal representation $r_{\text{st}} \in \mathbb{R}^d$. **Spectral Features:** In parallel to the spatiotemporal pathway, we incorporate a multi-scale spectral embedder to capture frequency-domain characteristics of EEG that are tightly linked to neural oscillatory dynamics. EEG contains meaningful information at multiple frequency bands (e.g., delta, theta, alpha, beta), and using a single spectral scale risks missing fine- or coarse-grained patterns. To address this, we compute spectral representations at multiple adaptive window sizes. Specifically, we apply the short-time Fourier transform (STFT) to $X$ at several scales. Given the base window scale $w_b$, at each level $l$ (with window $w_l = w_b \times 2^l$), we obtain patches $x_{l,c,k} \in \mathbb{R}^{w_l}$ and compute their FFT to yield spectra $s_{l,c,k} \in \mathbb{R}^{\frac{w_l}{2}+1}$ with $k = 1, ..., \lfloor T/w_l \rfloor$. These multi-level spectra are processed through frequency and temporal embedding modules, generating window embeddings with same shape $we_l$, which are then integrated to form the final spectral embedding: $e_{\text{sp}} = \sum_{l=0}^{L} we_l \in \mathbb{R}^{C \times n \times d}$. For each EEG channel, we apply positional embeddings, followed by a linear Transformer encoder (Katharopoulos et al., 2020) and average pooling operation, producing the final spectral representation: $r_{\text{sp}} \in \mathbb{R}^d$.

**Gated Adaptive Fusion (GAF).** We introduce GAF module to integrate the complementary features $r_{\text{st}}, r_{\text{sp}} \in \mathbb{R}^d$ derived from EEG encoder. The fused representation is

computed as: $r_{\text{fused}} = g \cdot r_{\text{st}} + (1 - g) \cdot r_{\text{sp}}$, where the gating factor $g$ is dynamically learned via a small multilayer perceptron (MLP). Two embeddings are first concatenated to derive $r_{\text{cat}} \in \mathbb{R}^{2d}$, and then processed by the MLP $g = \sigma\Big(W_2 \, \text{GELU}\big(W_1 \, r_{\text{cat}}\big)\Big)$, where $W_1$ and $W_2$ are learnable weight matrices and $\sigma$ is the sigmoid activation function. This adaptive gating mechanism emphasizes the most informative features from each dimension.

**Learnable ROI Lookup Table (LROI-LUT).** LROI-LUT is a trainable embedding that captures region-specific features while incorporating global EEG context. The LROI-LUT consists of $P$ learnable embeddings with random initialization, each corresponding to a predefined ROI. During training, the global EEG representation $r_{\text{fused}}$ is concatenated with each ROI embedding $r_i \in \mathbb{R}^{d_r}$ from the lookup table: $e_i = \big[r_{\text{fused}}; r_i\big]$, for all $i \in \{1, \ldots, P\}$, conditioning each ROI on EEG signal adaptively. The concatenated embedding $e_i$ is then passed to the RRE (described next) to transform the EEG-domain information into ROI-aware representation.

**ROI Representation Embedder (RRE).** RRE is a lightweight LoRA-like adapter (Hu et al., 2022; Wang et al., 2024) that transforms EEG-informed embeddings into a space aligned with fMRI representations. It compresses high-dimensional features into a lower-dimensional bottleneck, then restores them to the original space, optionally incorporating a residual connection for stability. Let $e \in \mathbb{R}^D$ denote an input concatenated embedding (i.e., one of the $e_i$). The process is as follows: **(1)** The input embedding is passed through a down-projection layer to obtain a bottleneck embedding $e_b = \text{ReLU}\Big(\text{Linear}_{\text{down}}(e)\Big)$. **(2)** After applying dropout for regularization, the bottleneck embedding is up-projected back to its original dimensionality using the transformation $\hat{e} = \text{Linear}_{\text{up}}(e_b)$. **(3)** A learnable scaling factor $\alpha$ modulates the up-projected features, and the final output is computed with a residual connection: $e_{\text{out}} = \alpha \cdot \hat{e} + e$. Finally, the generated ROI-specific refined embeddings are then processed by a set of **ROI-specific linear prediction heads** to predict the final fMRI signal.

**Multi-Objective Loss.** For predicting multi-region signals and reconstructing FC, capturing inter-regional interactions is crucial, as reconstructing individual ROIs may overlook these dependencies. To address this, we enforce biologically plausible constraints by integrating mean squared error (MSE), temporal correlation loss $\mathcal{L}_{\text{tcorr}}$, and spatial correlation loss $\mathcal{L}_{\text{scorr}}$ into a single objective:

$$\mathcal{L}_{\text{MO}} = \alpha \, \mathcal{L}_{\text{MSE}} + \beta \left(\frac{1}{P}\sum_{p=1}^{P}\Big(1 - R_{\text{t},p}\Big)\right) + (1 - \alpha - \beta)\left(\frac{1}{B}\sum_{b=1}^{B}\Big(1 - R_{\text{s},b}\Big)\right),$$

where $\alpha$ and $\beta$ control the contributions of each term, $R_{\text{t},p}$ is the Pearson correlation coefficient computed along the temporal axis (per ROI) for the $p$th time series, and $R_{\text{s},b}$ is computed along the spatial axis (per time point) for the $b$th batch element. During training, time points in each batch remain in their original order to compute the temporal loss, while the batch order is shuffled to prevent overfitting to sequential patterns.

## 3. Experiments and Analysis

### 3.1. Dataset and Experimental Settings

**Dataset**  We conducted our experiments on the shared resting-state EEG-fMRI dataset from (Li et al., 2024b), following the same pre-processing pipelines. This dataset comprises 29 simultaneous EEG-fMRI scans from 22 healthy volunteers in an eye-closed resting-state with 7 participants having two scans. Each scan lasts 20 minutes. Scalp EEG was recorded using a 32-channel MR-compatible system (10-20 system), with 26 channels retained after excluding ECG, EOG, and EMG channels. Dictionaries of Functional Modes atlas (Dadi et al., 2020) with n=64, 128, 256 ROIs are used to extract fMRI signals. We extract a 16-second EEG window, which is resampled to 200Hz, prior to each fMRI time point as the model's input for fMRI frame prediction. Further details of the EEG and fMRI preprocessing steps can be found in the original NeuroBOLT paper (Li et al., 2024b).

**Implementation Details**  We conduct our experiments using PyTorch 2.0.1 (Paszke et al., 2019) and Python 3.9.12 on a single NVIDIA RTX A5000 GPU (CUDA 11.8) with batch sizes of 16 for intra-scan and 64 for cross-subject analyses, training for 20 epochs using AdamW (initial learning rate $3 \times 10^{-4}$, weight decay 0.05, minimal learning rate $1 \times 10^{-6}$). The EEG embedding and RRE bottleneck dimensions are 200 and 128, respectively, with $d_r$ equal to the number of ROIs, and MO loss weights $\alpha = 0.8$ and $\beta = 0.1$. FC reconstruction metrics are computed on the upper (or lower) triangular portion of the matrix, excluding the diagonal. We follow the data partitioning in (Li et al., 2024b): an 8:1:1 split with 20-second gaps for intrascan predictions and a 3:1:1 ratio for cross-subject analysis. For subjects with multiple scans, all scans from the same individual are assigned to the same split to avoid information leakage during the cross-subject analysis. The EEG encoder's spatiotemporal module is initialized with the pretrained LaBraM-base model (Jiang et al., 2024) (token length 200, 1 second without overlap), and the multi-scale spectral module uses a smallest scale size $l_0 = 100$ (0.5 seconds without overlap). We use official released codebases for previous models for fair comparison. Our code will be released upon acceptance.

### 3.2. Results

We conduct experiments in two scenarios: **intrascan prediction**, where we train and test our model on the first 16 minutes and the last 2 minutes of the same scan, respectively; and **unseen subject prediction**, where the model is trained on a training set (18 scans) and evaluated on 6 held-out scans for full-scan (20 min) reconstruction. We evaluated the model's performance in two key aspects: (1) time course prediction, quantified as the temporal correlation ($R$) between predicted and real signals; (2) FC reconstruction, assessed by comparing FC metrics derived from the generated and true signals, including pixel-wise correlation (Pixcorr), connectivity MSE (ConnMSE) to measure structural deviations in network topology, and F1 scores for edge detection, measuring how well the model identifies the top 25% and 50% of the strongest connections.

#### 3.2.1. Performance Comparison

We compare our model with the SOTA baselines in EEG encoding (Jiang et al., 2024; Yang et al., 2023; Li et al., 2022; Song et al., 2021; Peh et al., 2022) and EEG-to-fMRI

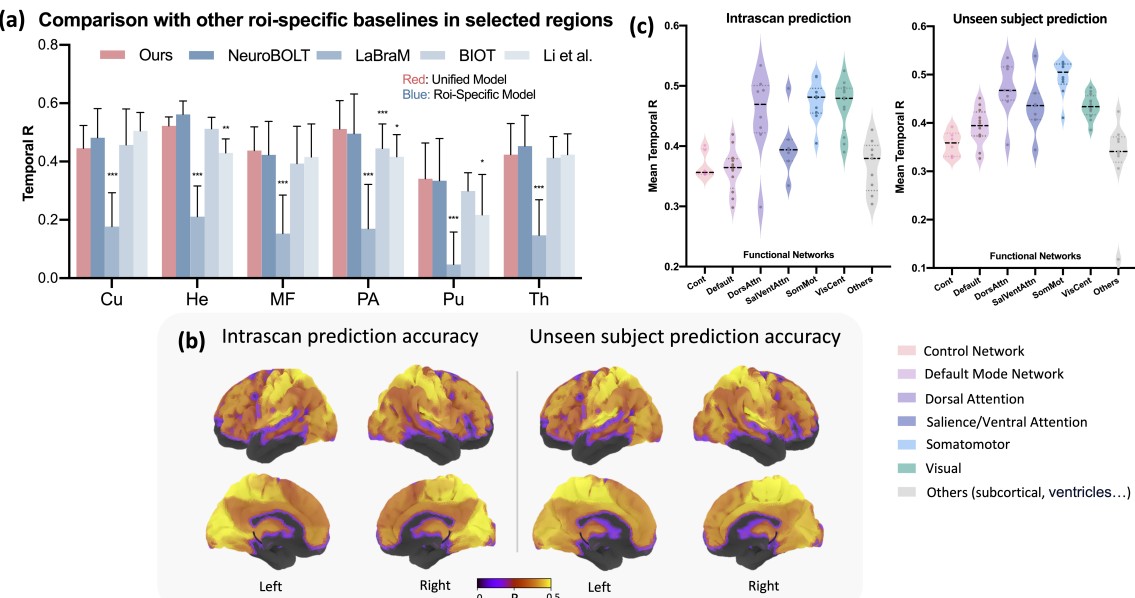

Figure 2: (a) Unseen subject prediction performance compared with ROI-specific baselines. [Bars: mean temporal $R$, error bars: S.D.; $*p < 0.05$, $**p < 0.01$, $***p < 0.001$: the paired t-test significance between our model and each baseline. Cu: cuneus, He: Heschl's gyrus, MF: middle frontal gyri, PA: anterior precuneus, Pu: Putamen, Th: Thalamus]. (b) Prediction accuracy in various brain regions. (c) Intrascan (left) and unseen subject (right) predictions. Each point represents the mean temporal $R$ across all scans for a given ROI.

translation (Li et al., 2024b,a; Kovalev et al., 2022). Since all baseline models are ROI-specific models (RSM), we first evaluate our unified model (UM) against RSM baselines in reconstructing fMRI signals across selected brain regions in (Li et al., 2024b), following the same benchmarking protocol. As shown in Fig.2(a), our framework, trained jointly on 64 brain regions within a single architecture, achieves competitive reconstruction performance while being significantly more efficient compared with iterative ROI-specific modeling.

To extend the comparison to a whole-brain analysis, we adapt the RSM baselines by modifying the final projection layer to map the latent embeddings to the entire set of ROIs instead of a single region, thereby constructing UM baselines. UnEBOLT outperforms all baselines in signal prediction and FC reconstruction across intrascan and unseen subject settings (top two section of Table 1). These results suggest that using a single final projection layer in the baseline backbone is not sufficient to capture dependencies among ROIs, highlighting the importance of a structured multi-region learning approach.

Fig.2 (b) shows the spatial distribution of prediction accuracy of our model. To further analyze the prediction accuracy across functional networks (Fig.2(c)), each ROI is mapped to its corresponding functional network, as defined by Yeo's 7-network parcellation (Yeo et al., 2011). The Somatomotor Network exhibits the highest prediction accuracy in both

Table 1: Merged comparison and ablation studies (P=64), and model performance across different numbers of ROIs, reported as Mean(std). **Bold**: the best mean value; gray: previous SOTA; blue: architecture ablation; green: loss ablation; yellow: our full model.

| | Method | Multi-Region | Tcorr | Pixcorr | ConnMSE | F1-0.25 | F1-0.5 |
|---|---|---|---|---|---|---|---|
| intra (P=64) | BIOT (Yang et al., 2023) | ✗ | 0.339(0.169) | 0.396(0.206) | 0.224(0.134) | 0.421(0.090) | 0.635(0.068) |
| | LaBraM (Jiang et al., 2024) | ✗ | 0.184(0.164) | 0.248(0.226) | 0.381(0.229) | 0.366(0.104) | 0.591(0.077) |
| | Li et al. (Li et al., 2024a) | ✗ | 0.391(0.195) | 0.451(0.224) | **0.154(0.101)** | 0.466(0.104) | 0.660(0.074) |
| | NeuroBOLT (Li et al., 2024b) | ✗ | 0.369(0.164) | 0.402(0.213) | 0.259(0.182) | 0.427(0.088) | 0.641(0.070) |
| | Ours (P=64) | ✓ | **0.416(0.166)** | **0.490(0.176)** | 0.201(0.148) | **0.474(0.067)** | **0.672(0.062)** |
| prediction (P=64) | ST-Transformer (Song et al., 2021) | ✗ | 0.107(0.077) | 0.295(0.132) | 0.244(0.025) | 0.368(0.062) | 0.593(0.057) |
| | CNN-Transformer (Peh et al., 2022) | ✗ | 0.105(0.096) | 0.226(0.113) | 0.281(0.043) | 0.343(0.040) | 0.576(0.060) |
| | FFCL (Li et al., 2022) | ✗ | 0.190(0.067) | 0.245(0.049) | 0.136(0.115) | 0.401(0.012) | 0.605(0.024) |
| | BIOT (Yang et al., 2023) | ✗ | 0.413(0.035) | 0.486(0.135) | 0.098(0.055) | 0.444(0.055) | 0.626(0.046) |
| | LaBraM (Jiang et al., 2024) | ✗ | 0.214(0.058) | 0.393(0.089) | 0.122(0.068) | 0.460(0.076) | 0.611(0.049) |
| | BEIRA (Kovalev et al., 2022) | ✗ | 0.148(0.102) | 0.389(0.159) | 0.293(0.055) | 0.380(0.067) | 0.597(0.048) |
| | Li et al (Li et al., 2024a) | ✗ | 0.370(0.047) | 0.483(0.109) | 0.080(0.047) | 0.472(0.056) | 0.642(0.053) |
| | NeuroBOLT (Li et al., 2024b) | ✗ | 0.413(0.052) | 0.428(0.145) | 0.126(0.080) | 0.475(0.057) | 0.637(0.053) |
| unseen subject | w/o ROI-specific head | ✓ | 0.407(0.043) | 0.217(0.093) | 0.201(0.093) | 0.333(0.029) | 0.576(0.036) |
| | w/o ROI embedder | ✓ | 0.417(0.050) | 0.484(0.097) | 0.096(0.069) | 0.533(0.054) | 0.667(0.041) |
| | w/o LROI-LUT | ✓ | 0.410(0.045) | 0.480(0.103) | 0.096(0.065) | 0.509(0.018) | 0.670(0.042) |
| | w/o GAF | ✓ | 0.407(0.049) | 0.541(0.095) | 0.072(0.055) | 0.544(0.067) | 0.684(0.043) |
| | only mse loss | ✓ | 0.400(0.029) | 0.464(0.098) | 0.099(0.069) | 0.513(0.035) | 0.668(0.033) |
| | w/o mse loss | ✓ | 0.392(0.071) | 0.510(0.113) | 0.086(0.059) | 0.496(0.068) | 0.668(0.043) |
| | w/o scorr loss | ✓ | 0.402(0.120) | 0.480(0.120) | 0.114(0.072) | 0.516(0.035) | 0.672(0.035) |
| | w/o tcorr loss | ✓ | 0.401(0.064) | 0.498(0.099) | **0.071(0.047)** | 0.520(0.050) | 0.678(0.037) |
| | Ours (P=64) | ✓ | **0.418(0.036)** | **0.549(0.120)** | 0.092(0.066) | **0.545(0.074)** | **0.690(0.047)** |
| | Ours (P=128) | ✓ | 0.386(0.044) | 0.560(0.113) | 0.103(0.071) | 0.516(0.050) | 0.678(0.038) |
| | Ours (P=256) | ✓ | 0.360(0.047) | 0.621(0.083) | 0.152(0.065) | 0.511(0.042) | 0.684(0.035) |

conditions, followed by the Dorsal Attention and Visual networks. Additionally, intrascan predictions exhibit greater individual variability, while unseen subject predictions achieve higher overall accuracy. This trend may be attributed to the shorter temporal window used for intrascan evaluation, which limits temporal coverage and constrains the range of brain states observed during training, leading to higher variability in the estimated representations. Perhaps as a result, model performance under this setting is more sensitive to the specific brain state during the testing interval. In contrast, unseen-subject prediction is evaluated on full-length scans, possibly yielding more stable and reliable estimates. Interestingly, the Visual Cortex exhibits higher predictability in the intrascan setting, suggesting that subject-specific training may better capture individual visual-process variations.

Beyond the ROI-wise analysis, we also examined whether the model can recover coherent dynamics at the network level, rather than only at the individual ROI level. For each functional network, we averaged the fMRI time series across all ROIs belonging to that network to obtain a representative network-level signal, and evaluated how well the network-level signal reconstructed from our model matches the corresponding ground-truth trajectory. This analysis allows us to assess whether UnEBOLT preserves mesoscale functional organization and whether the predicted regional signals collectively recover higher-level network dynamics. Fig. 3 summarizes the network-level reconstruction performance. Panel (a) shows the distribution of reconstruction accuracy across networks for unseen scans, while panel (b) presents example time-series visualizations, each depicting the best prediction instance

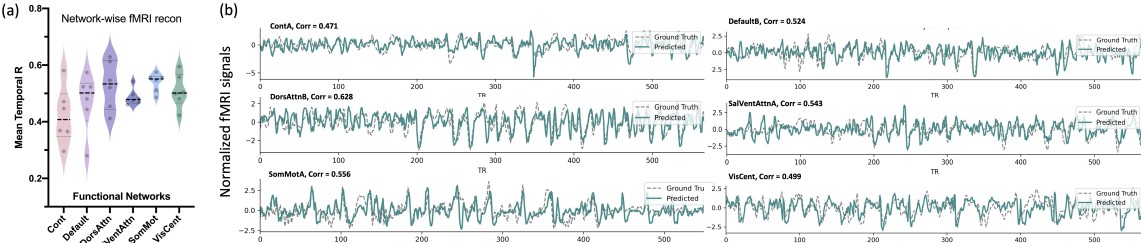

Figure 3: Network-wise fMRI time-series reconstruction performance on unseen scans. (a) Prediction performance across functional networks; each dot corresponds to an individual unseen scan. (b) Representative reconstruction examples, with each subplot showing one example drawn from each network.

within a given network. Consistent with our ROI-wise findings, UnEBOLT accurately recovers temporal fluctuations at the network scale, demonstrating that the model captures not only localized region-level patterns but also their coordinated network-level interactions. Across networks, the Somatomotor (SomMot) network again exhibits the highest reconstruction accuracy, followed by the Dorsal Attention and Visual networks. This ordering closely parallels the ROI-wise results in Fig. 2, and is consistent with prior EEG–fMRI studies showing that sensory–motor and visual systems display some of the strongest electrophysiology–hemodynamic coupling during rest (Xavier et al., 2025). Overall, these findings highlight that UnEBOLT learns representations that generalize from fine-grained ROI predictions to coherent, mesoscale network dynamics.

The final section of Table 1 evaluates the model's performance in predicting unseen subjects' fMRI data across different levels of spatial granularity (i.e., varying numbers of ROIs; example visualization: Fig. 4). As illustrated in Fig. 4, the recovered FC matrices closely resemble the ground truth across different parcellation levels and sparsity thresholds. For coarser parcellations (P=64), the block structure of canonical functional networks is clearly recovered, yielding high correspondence with an F1 score up to 0.77 at the 0.50 threshold. Predicting time series for a larger number of ROIs is more challenging, as reflected in decreased Tcorr, ConnMSE, and F1 for the 0.25 connectivity threshold, likely due to increased complexity and reduced signal-to-noise ratio in finer parcellations. Interestingly, however, higher ROI resolution improves the spatial structure prediction of FC, as indicated by increasing Pixcorr.

### 3.2.2. Ablation Studies

Compared with ablated variants, our full model achieves the highest performance across most metrics (Table 1). Removing the ROI-specific projection head significantly degrades performance, highlighting its role in capturing region-specific information. Excluding the ROI embedder also substantially reduces Pixcorr, indicating its importance in maintaining FC structure. While the model can still capture temporal dynamics, the drop in Pixcorr suggests the embedder refines spatial organization. Omitting spatial loss further weakens ROIs' spatial relationships. Overall, these findings underscore the necessity of each com-

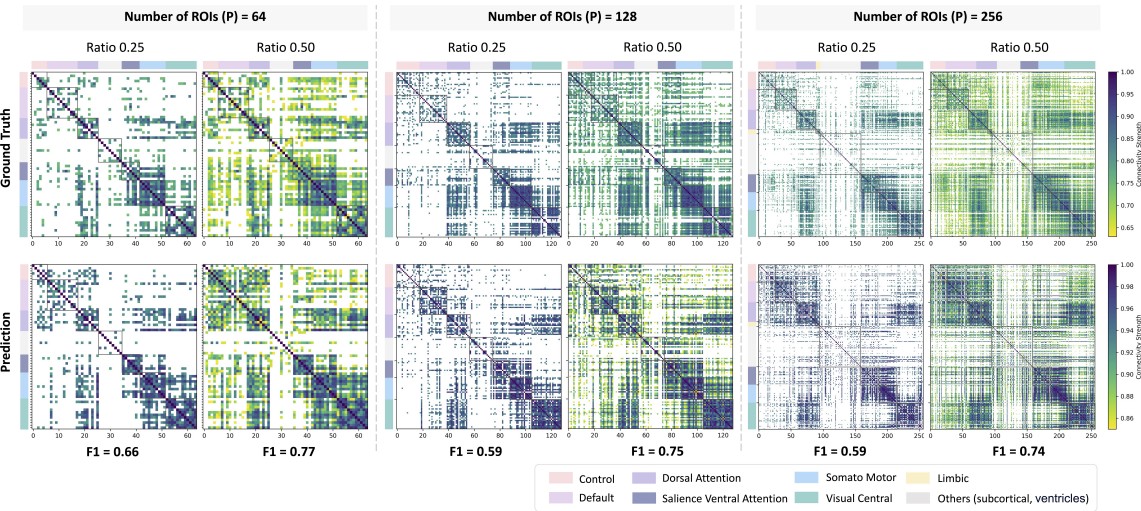

Figure 4: Examples of whole-scan FC reconstruction from predictions on an unseen subject across varying ROI resolutions and sparsity ratios. Top row: ground-truth FC; bottom row: predicted FC. ROIs are grouped by functional network and ordered contiguously within each group, with network membership indicated by the accompanying color bars.

ponent in enabling UnEBOLT to jointly model regional specificity and cortical structure, ultimately supporting accurate time-series prediction and FC recovery.

### 3.2.3. Model Generalization Analysis

The present study primarily focuses on resting-state fMRI synthesis. To further assess the generalizability of the pretrained model to data acquired under different experimental conditions, we additionally evaluate the model on an auditory task-based EEG–fMRI dataset introduced in (Li et al., 2024b). This dataset was collected at different sites using distinct acquisition devices, providing a challenging evaluation setting. During the scans, binaural auditory stimuli were presented with randomized inter-stimulus intervals (ISI), and participants were instructed to press a button as soon as they heard each stimulus.

The task-based dataset comprises 16 scans from 10 healthy subjects, including 9 training scans, 3 validation scans, and 4 test scans. For a detailed description of the dataset and experimental protocol, we refer the reader to (Li et al., 2024b). Using this dataset, we conduct three types of unseen-subject scan prediction experiments: (1) Zero-shot transfer, in which the model is trained exclusively on resting-state data and directly evaluated on task-condition data; and (2) Fine-tuning, where the pretrained model is further fine-tuned on the task-condition training set prior to evaluation. (3) Training from scratch using task-condition data.

Due to differences in EEG channel configurations between the two datasets, we use only the 23 EEG channels that are shared across both datasets as model input. As shown in Table 2, the model demonstrates strong generalization to data collected under a different

experimental paradigm and with different acquisition hardware, despite using only a subset of the channels available in the full model configuration. Moreover, fine-tuning on the task-specific training data further improves performance across evaluation metrics.

Table 2: Task-condition unseen scan reconstruction performance (P=64), reported as Mean(std) with best mean values in **Bold**.

| Approach | Pretrain | Tcorr | Pixcorr | ConnMSE | F1-0.25 | F1-0.50 |
|---|---|---|---|---|---|---|
| Zero-shot | ✓ | 0.377(0.023) | 0.352(0.148) | 0.093(0.040) | **0.480**(0.070) | 0.656(0.048) |
| Fine-tune | ✓ | **0.406**(0.083) | **0.421**(0.220) | **0.086**(0.030) | 0.472(0.052) | **0.666**(0.095) |
| From scratch | ✗ | 0.379(0.071) | 0.352(0.237) | 0.098(0.028) | 0.386(0.041) | 0.628(0.088) |

## 4. Discussion and Conclusion

We propose UnEBOLT, a **Un**ified, **E**EG-to-**BOL**D **T**ranslation model designed for efficient end-to-end reconstruction of fMRI signals and functional connectivity from EEG. Unlike existing ROI-specific models, UnEBOLT leverages an adaptive ROI-specific representation learning mechanism in a multi-region joint learning framework, improving training efficiency and scalability. In our experiments with the eyes-closed resting-state dataset, UnEBOLT demonstrates superior performance for whole-brain fMRI time series prediction and functional connectivity reconstruction in both intrascan and unseen subject settings compared with baselines. Furthermore, the model exhibits promising generalization capability when transferred to an unseen task-condition dataset. While the current study primarily focuses on eyes-closed resting-state EEG-fMRI synthesis, future work will extend the evaluation to a broader range of task-based paradigms as additional task-based EEG-fMRI datasets become available.

The strength of agreement between our predictions and ground-truth fMRI signals underscores UnEBOLT's reliability and highlights its promise as a cost-effective, end-to-end tool for inferring fMRI from EEG. This unified approach opens new possibilities for large-scale brain research and future downstream applications in cognitive neuroscience, clinical diagnostics, and brain–computer interfaces.

## Acknowledgments

We are grateful to support from the Sally and Dave Hopkins Faculty Fellowship and from NIH grants R01NS112252, P50MH109429, T32EB021937, and F31NS143413.

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

## Appendix A. Subject-wise Results

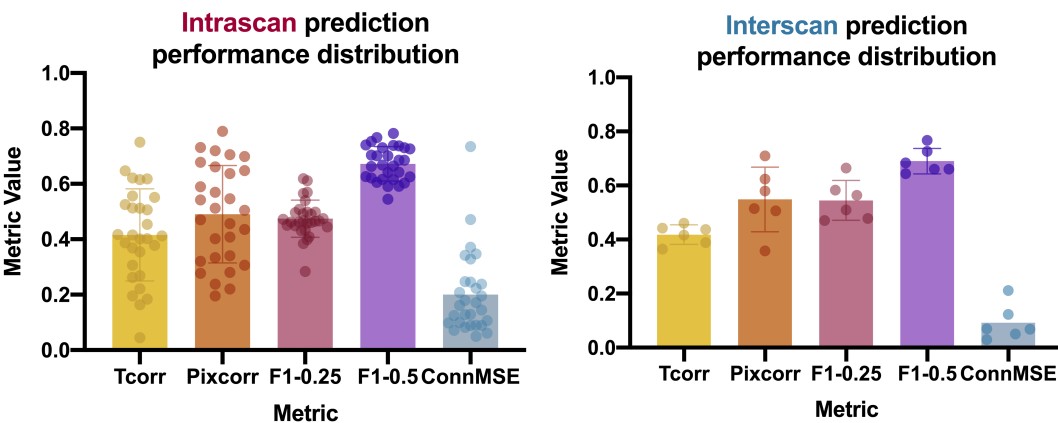

Figure 5: Intrascan (left) and unseen subject (right) prediction performance distribution. Bars: mean values; Error bars: S.D.. Each point represents each scan.

Table 3: Subject-level performance (P=64) in unseen subject fMRI synthesis (N=6), reported as Mean(std).

| Subject ID | Tcorr | Pixcorr | ConnMSE | F1-0.25 | F1-0.50 |
|---|---|---|---|---|---|
| sub07-scan01 | 0.460 | 0.506 | 0.068 | 0.470 | 0.661 |
| sub07-scan02 | 0.416 | 0.358 | 0.123 | 0.478 | 0.644 |
| sub11-scan01 | 0.437 | 0.710 | 0.050 | 0.665 | 0.767 |
| sub16-scan01 | 0.442 | 0.515 | 0.211 | 0.510 | 0.661 |
| sub18-scan01 | 0.365 | 0.623 | 0.029 | 0.583 | 0.684 |
| sub21-scan01 | 0.389 | 0.579 | 0.069 | 0.563 | 0.726 |
| **Mean(std)** | **0.418(0.036)** | **0.549(0.120)** | **0.092(0.066)** | **0.545(0.074)** | **0.690(0.047)** |

