# OpenReview forum: "UnEBOLT: A Unified Model for EEG-to-BOLD Translation and Functional Connectivity Reconstruction"
_MIDL.io/2026/Conference — MIDL 2026 Poster_

### Official Review · Reviewer_CNZo · 2026-01-08

**Confidence:** 4
**Preliminary Rating:** 4
**Final Rating:** 4

**Summary:**

This paper introduces UnEBOLT, a unified deep learning framework for EEG-to-BOLD fMRI translation that enables whole-brain fMRI time-series prediction within a single model, addressing the inefficiency of ROI-specific approaches. The method combines multi-dimensional EEG encoding, gated adaptive fusion, and ROI-specific representation learning, and is trained using a multi-objective loss that enforces reconstruction accuracy as well as temporal and spatial correlations. Experiments on resting-state EEG–fMRI data demonstrate competitive performance against multiple baselines in both intrascan and unseen-subject settings, while also enabling reconstruction of subject-specific functional connectivity.

**Strengths:**

1. Introduced a novel and scalble method for EEG-to-fMRI translation across multiple brain regions within a single architecture, directly addressing the inefficiency of training separate models per ROI that dominates prior work
2. Presented extansive experiments and evaluation including intrascan and unseen subject prediction, the performance of the proposed method is compared against multiple baselines.
3. Ablation study is conducted and clearly demonstrated the contribution and effectiveness of the ROI-specific projection and representation module.

**Weaknesses:**

1. The potential of the learned ROI lookup table is underexplored. While ablation results demonstrate its effectiveness, the paper does not analyze whether these learned embeddings capture meaningful anatomical, functional, or network-level information, which could provide deeper neuroscientific insight and improve interpretability.
2. Experimental validation is limited to an eyes-closed resting-state dataset. It remains unclear how well the proposed framework would generalize to other settings, such as eyes-open resting state or task-based EEG–fMRI data, which are common in both cognitive and clinical studies.

**Detailed Comments:**

1. The paper would benefit from a clearer description of the intended role of the learnable ROI lookup table beyond serving as an index for region-specific conditioning. For example if the embebdding encodes any anatomical identity, functional specialization.

**Justification Of Final Rating:**

The author address my concerns in the rebuttal, I would like to raise my score to 5. This paper introduces UnEBOLT, a unified deep learning framework for EEG-to-BOLD fMRI translation that enables whole-brain fMRI time-series prediction within a single model, addressing the inefficiency of ROI-specific approaches.

**Justification Of The Preliminary Rating:**

The paper presents a novel and well-motivated unified framework for EEG-to-fMRI translation that addresses a key scalability limitation of prior ROI-specific methods. The proposed method is supported by extensive experimental evaluations, including multi-region prediction, unseen-subject generalization, and comparisons with multiple strong baselines, demonstrating its technical soundness and practical potential. However, several aspects would benefit from further clarification, particularly regarding the interpretability of the learned ROI embeddings.

**Questions To Address In The Rebuttal:**

1. Can the authors provide additional analysis or intuition on what the learned ROI lookup table embeddings represent? Or are these just served as a different "index" for region-specific conditioning?
2. Does the method generalize beyond eyes-closed resting state?

---

> ### Author Response · Authors · 2026-01-25
>
> We sincerely thank the reviewer for the insightful and constructive feedback. Below, we provide a detailed, point-by-point response to address the concerns raised.
>
> ---
>
> >**Exploration on learned ROI lookup table embedding (W1, Q1)**
>
> We thank the reviewer for this thoughtful suggestion. We agree that the learned ROI lookup table has the potential to provide additional neuroscientific insight beyond its role in conditioning. To this end, we conducted preliminary analyses by visualizing the learned ROI embeddings and examining pairwise similarities among ROI-specific embeddings. However, in our current experiments, we did not observe clear or consistent anatomical, functional, or network-level patterns (e.g., alignment with known functional connectivity structures). We believe this is likely due to the limited sample size and the relatively weak supervision signal available for disentangling fine-grained neurobiological structure at the level of ROI embeddings. At the current stage, the ROI lookup table primarily serves as a learnable indexing and conditioning mechanism that enables region-specific modulation within a unified model, rather than as an explicitly interpretable representation.
>
> We view a more in-depth analysis of the learned ROI embeddings, potentially leveraging larger datasets, stronger regularization, or explicit neurobiological priors, as an important direction for future work.
>
> ---
>
> >**Model generalization to other conditions (W2, Q2)**
>
> Thank you for your valuable comment. To evaluate how well our model trained on a resting-state dataset transfers across task conditions and scanners,  we conduct experiments on **an additional auditory task-based simultaneous EEG-fMRI dataset** collected at a different site using a different scanner. Due to differences in EEG channel configurations between the two datasets, we use only the 23 EEG channels that are shared across both datasets as the model input for evaluation.
>
> Using this dataset, we evaluate the model under three settings:
> (1) **zero-shot prediction,** where the pretrained model is frozen and directly evaluated on task-condition data;
> (2) **fine-tuning**, where the pretrained model is further adapted using task-condition training data; and
> (3) **training from scratch on the task-condition dataset.**
>
> Detailed experimental settings and results are provided in ***Section 3.2.3 (Model Generalization Analysis)*** of the **revised manuscript**. As shown in ***Table 2***, the model pretrained on resting-state data achieves competitive zero-shot performance comparable to models trained within the task domain, demonstrating robust generalization across different experimental conditions and acquisition devices. These results highlight the potential of the proposed model for broader application to task-based EEG-fMRI datasets.

---

### Official Review · Reviewer_bS6G · 2026-01-09

**Confidence:** 5
**Preliminary Rating:** 4
**Final Rating:** 4

**Summary:**

The authors propose UnEBOLT, as an end-to-end framework that predicts ROI-level fMRI time series from simultaneous EEG within a single model, and then uses the generated time series to reconstruct subject-specific functional connectivity (FC) for validation. The model combines a spatio-temporal EEG encoder with a gated adaptive fusion module and a learnable ROI lookup together with ROI-specific heads, trained with a multi-objective loss that includes temporal or spatial correlation constraints. The authors applied the proposed network on eyes-closed EEG recordings and show strong performance with extensive comparisons and ablations.

**Strengths:**

The paper has a clear motivation that directly addresses the inefficiency of ROI-specific EEG to fMRI models and formalizes a scalable alternative that predicts all ROIs in one pass. This design saves the computation overhead from the previous model.

The design cleanly separates the EEG representations of spatial-temporal characteristics, adaptive fusion via a learned gate, and the ROI-conditioned decoding via a learnable ROI embedding table and adapter module. These designs are justified by solid evaluation and ablations in the paper with both time-series and FC reconstruction metrics.

Training time is reported, and the unified approach is described as substantially faster than training per-ROI models. This strengthens their claim on streamlining and redesigning previous models.

**Weaknesses:**

Experiments use a relatively small eyes-closed resting-state dataset (29 scans on 22 volunteers). It’s unclear how robust the approach is to different scanners, montages, preprocessing pipelines, or task/eyes-open conditions.

It is not clearly demonstrated why the authors chose to transform the EEG into fMRI data. EEG has limited spatial resolution, while fMRI has higher spatial resolution. The authors already showed that by using ROI aggregation, fMRI spatial resolution is reduced so that EEG could be used to generate the ROI time series network dynamics could be reconstructed. However, the authors did not discuss any downstream tasks that could benefit from this model. For example, could any patients without the fMRI data benefit from a pre-trained fMRI neurological disease classifier? Will these data potentially boost the classification accuracy?

**Detailed Comments:**

Some presentation or clarity issues. There appears to be a figure-reference mismatch (“Fig.1(a)” while discussing the ROI-specific baseline comparison that is shown in Fig. 2(a)), which can confuse readers. Also, in "ROI Representation Embedder (RRE)" under the Methods section, when describing the second process, there should be a space separating "transformation" and "e\hat" notation.

**Justification Of Final Rating:**

The author proposes UnEBOLT to predict whole brain ROI time sereis in one pass from EEG. The rebuttal addressed well about the generalization by adding an evaluation on the auditory task dataset from a different site, which strenghtes the claim that the learned mapping can transfer across acquisition conditions. While the author addressed the motivation better and is more clear, the paper still does not demonstrate a concrete downstrem benefit and the external validation is limited. Therefore, I would like to remain my rating.

**Justification Of The Preliminary Rating:**

I recommend weak acceptance because the paper is well-motivated and addresses a clear practical bottleneck in EEG-to-fMRI modeling, and improves the computational inefficiency and scalability. The proposed unified framework predicts all ROIs in one pass, reducing the training overhead and offering a more deployable alternative while maintaining competitive performance.

Methodologically, the design is clean and well-structured. The design captures EEG spatiotemporal characteristics, performs adaptive fusion through a learned gating mechanism, and uses an ROI-conditioned decoder with a learnable ROI embedding table and adapter module. These choices are supported by solid experimental results and ablations across both time-series reconstruction and functional connectivity reconstruction metrics, making the claimed contributions credible. In addition, the paper reports training time and demonstrates that the unified approach is substantially faster than training per-ROI models, strengthening the argument that the proposed redesign meaningfully streamlines prior work.

I remain weak rather than strong due to two main limitations. First, the empirical evaluation is based on a relatively small eyes-closed resting-state dataset (29 scans from 22 volunteers), so it is uncertain how robust the approach will be across different scanners, EEG montages, preprocessing pipelines, or task/eyes-open conditions. Second, while the EEG to fMRI translation is technically justified via ROI aggregation, the paper does not clearly articulate why generating fMRI-like signals is the most valuable objective, nor does it demonstrate downstream utility. In particular, it would strengthen the value and impact of this work to discuss or test whether predicted ROI fMRI time series can meaningfully support downstream applications (for example, improving neurological disease classification when fMRI is unavailable, transferring pretrained fMRI classifiers, or boosting prediction accuracy via multimodal augmentation). Overall, the paper presents a strong and scalable modeling direction with convincing core results, but would benefit from broader validation and clearer downstream motivation.

**Questions To Address In The Rebuttal:**

Subject-specific vs cross-subject clarification: The paper distinguishes intrascan (subject-specific) vs unseen-subject prediction. You note that unseen-subject achieves higher overall accuracy (likely due to more training data), while intrascan shows higher variability, and the visual cortex behaves differently. Could you expand on why subject-specific training underperforms here, and whether controlling training-set size changes this conclusion?

Figure 2A concern: Fig. 2(a) shows unseen-subject performance vs ROI-specific baselines in selected regions. For ROIs where the proposed model appears weaker than certain baselines, can you explain when/why this happens (for example: negative transfer across ROIs, insufficient region-specific capacity), and whether this trend persists across more ROIs than the displayed subset?

Robustness to parcellation choices and ROI granularity: Performance changes with different choices of P. Do you expect similar trends for different atlases/parcellations (not just different P), and can you comment on practical guidance for choosing P?

---

> ### Author Response · Authors · 2026-01-25
> **Author Rebuttal - Part 1**
>
> We sincerely appreciate the reviewer for the insightful and constructive review. Please find our detailed, point-by-point response to address your concerns below:
>
> ---
>
> >**Model generalization to the task-condition dataset (W1)**
>
> Thank you for your valuable comment. To evaluate how well our model trained on a relatively small resting-state dataset transfers across task conditions and acquisition settings, we conduct additional experiments on an additional auditory task-based simultaneous EEG-fMRI dataset collected at a different site using a different scanner. Due to differences in EEG channel configurations between the two datasets, we use only the 23 EEG channels that are shared across both datasets as the model input for evaluation. Apart from MRI scanner and ballistocardiogram artifact removal, only minimal EEG preprocessing is applied, consisting of resampling and rescaling.
>
> Using this dataset, we evaluate the model under three settings:
> (1) **zero-shot prediction**, where the pretrained model is frozen and directly evaluated on task-condition data;
> (2) **fine-tuning**, where the pretrained model is further adapted using task-condition training data; and
> (3) **training from scratch on the task-condition dataset**.
>
> Detailed experimental settings and results are provided in ***Section 3.2.3 (Model Generalization Analysis)*** of the **revised manuscript**. As shown in ***Table 2***, the model pretrained on resting-state data achieves competitive zero-shot performance comparable to models trained within the task domain, demonstrating robust generalization across different experimental conditions and acquisition devices. These results highlight the potential of the proposed model for broader application to task-based EEG-fMRI datasets.
>
> To summarize, despite the limited availability of public resting-state EEG-fMRI datasets, which constrains large-scale evaluation, we rigorously validate our approach on the current dataset and demonstrate consistent within-domain performance and encouraging out-of-domain generalization even with this smaller sample. A more comprehensive evaluation with a larger sample size and across diverse task paradigms is left for future work, as additional simultaneous EEG-fMRI datasets become publicly available.
>
> ---
>
> >**Benefits and rationale of transforming EEG into fMRI ROI data. (W2)**
>
> Thank you for your insightful question. In this work, we focus on ROI-level representations (e.g., ROI time series and functional connectivity) as a starting point because they offer improved signal-to-noise ratio compared to voxel-level signals, are widely adopted in fMRI research, and support more efficient and interpretable modeling. Although these representations are derived features, they preserve critical information enabled by fMRI’s whole-brain sensitivity and coverage \- capabilities that are not directly accessible from EEG alone.
>
> In particular, fMRI-derived features provide access to deep and subcortical brain regions whose activity and connectivity are difficult to reliably capture with EEG due to volume conduction effects, geometry, and distance from the scalp. By learning to map EEG signals into this fMRI-informed representational space, our model enables the estimation of large-scale network dynamics and connectivity patterns that include deeper cortical and subcortical regions. This, in turn, would support and improve future downstream analyses, such as disease classification or biomarker discovery, using EEG-only data at inference time, which is particularly valuable for patient populations for whom fMRI acquisition is unavailable, contraindicated, or impractical (e.g., due to metal implants, claustrophobia, or limited clinical accessibility). While a systematic evaluation of downstream clinical tasks is beyond the scope of the current work and limited by the availability of paired EEG-fMRI datasets, especially for patient populations, the proposed framework establishes a foundation for such applications.
>
> ---
>
> >**Presentation or clarity issues**
>
> Thank you very much for pointing this out\! These issues have been corrected in the revised manuscript and highlighted in purple.
>
> ---
>
> *Due to character limits, we continue our response in the next comment.*

---

> > ### Author Response · Authors · 2026-01-25
> > **Author Rebuttal - Part 2**
> >
> > >**Subject-specific vs cross-subject clarification (Q1)**
> >
> > Thank you very much for the insightful question. There are several factors that may contribute to the lower performance metrics observed under subject-specific training.
> >
> > First, the two settings differ in both evaluation protocols and data characteristics: unseen-subject prediction is evaluated on full-length scans, which yield more stable estimates, whereas intrascan prediction is assessed on much shorter held-out segments and is therefore inherently more variable. In addition, resting-state dynamics within a single scan can be highly nonstationary, and subject-specific training may not observe the full range of brain states encountered during evaluation.  Finally, as noted in the manuscript, more training data with more individual variability may also help the model generalize to unseen subjects. That said, certain regions may still benefit from subject-specific modeling due to pronounced inter-individual variability.
> >
> > In summary, the quantitative results from the two conditions (intra-scan and unseen subject prediction) may not be directly comparable due to these different settings, and we have revised the corresponding discussion to clarify that these factors jointly contribute to the observed differences, rather than attributing them solely to training set size.
> >
> >
> > ---
> >
> > >**Figure 2A concern (Q2)**
> >
> > Thank you for the question. While the observed differences between the unified model (UM) and ROI-specific models are not statistically significant, mean performance trends suggest that sensory/relay-related regions (e.g., cuneus, Heschl’s gyrus, and thalamus) may benefit slightly from ROI-specific specialization, whereas several higher-order association regions (e.g., middle frontal gyrus and anterior precuneus) tend to benefit more from joint training. One plausible explanation is that ROI-specific modeling can better capture relatively localized mappings for sensory/relay-related ROIs, while joint training provides a shared representation and multi-task regularization that is more advantageous for distributed, integrative dynamics in association cortex. That said, a more rigorous analysis with larger datasets and targeted statistical testing will be required to validate these observations. In future work, we also plan to enhance the unified model to jointly optimize computational efficiency and predictive accuracy.
> >
> > ---
> >
> > >**Robustness to parcellation choices and ROI granularity (Q3)**
> >
> > Thank you for your insightful questions\! Yes, we expect the observed trends to generalize to other commonly used atlases and parcellation schemes. The underlying behavior is primarily driven by ROI granularity and signal-to-noise trade-offs, rather than atlas-specific properties.
> >
> > From a practical perspective, our current results suggest that moderate ROI granularity (e.g., P=64) is overall preferable when the goal is robust temporal modeling and reliable network-level functional connectivity analysis, due to improved signal-to-noise ratio. While this study mainly focuses on reconstruction fidelity and functional connectivity structure, an important and interesting future direction is to systematically evaluate how different parcellation choices affect downstream tasks (e.g., disease classification, subject-level phenotyping) using the reconstructed fMRI signals when larger-scale labeled datasets become available. This would enable a more task-driven selection of ROI granularity, potentially favoring finer parcellations when spatial specificity is critical, or coarser parcellations when robustness and generalization are prioritized, and this would be our future work.

---

### Official Review · Reviewer_qSND · 2026-01-10

**Confidence:** 4
**Preliminary Rating:** 5
**Final Rating:** 5

**Summary:**

UnEBOLT suggests an EEG-fMRI translation model that utilizes whole-brain dynamics. Eyes-closed resting-state EEG signal is encoded by spatiotemporal (local component, initialized by LaBraM-base) and spectral (global component) pathways (adopted from NeuroBOLT) and the output of these pathways are fused into a region-aware embedding, implemented with a learnable lookup table. The results are evaluated in two ways; measuring temporal correlation and comparing functional connectivity between DiFuMo inferred fine-grained ROIs evaluated under Yeo's 7 network parcellation between real and predicted signals against recent ROI-specific models.
UnEBOLT achieves close reconstruction performance with a significantly faster runtime than the previous work.

**Strengths:**

The unified whole brain model is significantly faster in running and inference time than the recent ROI-based models.

The model operates on the whole-brain signal yet has competitive performance in the reconstruction of region-specific metrics compared to ROI-based models.

The writing is well-structured and easy to follow.

Ablation study involves all model components and loss terms, where only in the case without temporal correlation loss term, functional connectivity mean squared error is significantly lower.

The model details are given with enough detail for reproducibility.

**Weaknesses:**

Time efficiency claim is supported by the advantage against ROI-specific models (RSM) iteratively run for each region. The runtime comparison requires more careful justification. The key advantage of the whole-brain unified model (UM) over iterative ROI models (like NeuroBOLT) is clear, but the fairness of this comparison should be discussed. A more informative baseline might involve comparing against region specific models with whole brain projections (pseudo-UM baselines in section 3.2.1). How do the time efficiency of pseudo-UM models change compared to iteratively running RSM for each region?

The paper would benefit from a clearer positioning within the literature. See the questions to address section.

The motivation behind choosing closed-eye rs-fMRI data could be further detailed in the paper. See the detailed comments section.

A table or plot of subject performances (one column/axis with subject IDs) could make the results more transparent regarding its stability.

**Detailed Comments:**

The use of closed-eye resting-state fMRI data is a significant limitation, as it precludes validation against known task-related neural invariants. Given the inherent uncertainties in EEG-BOLD translation, grounding the findings in established task-based neuroscientific outcomes would substantially strengthen the work.

**Justification Of Final Rating:**

The authors have provided a thorough and compelling point-by-point revision that directly addresses all my concerns, particularly strengthening the manuscript's claims, contextualization, and validation. My initial strong support for acceptance is firmly upheld.

**Justification Of The Preliminary Rating:**

The article is well structured and written. The study evaluates multiple datasets. The findings are convincing with experimental depth, supported by ablations. The suggested improvement is a good fit for the conference audience.

**Questions To Address In The Rebuttal:**

The paper would benefit from a clearer positioning within the literature. The article neither explicitly claim to be the first unified whole-brain EEG-fMRI translation model nor compares against any other (originally) whole-brain EEG encoding or EEG-fMRI model in the evaluation. Can you clarify this point?

---

> ### Author Response · Authors · 2026-01-25
>
> We sincerely thank the reviewer for the insightful comments and encouraging feedback on our manuscript. We have provided a detailed, point-by-point response to address your concerns:
>
> ---
>
> >**More careful efficiency claim (W1)**
>
> Thank you very much for pointing this out. We agree that the runtime comparison requires more careful justification. The pseudo-UM model we built for the baseline comparison is a single-layer MLP attached to the shared EEG embedding with the output dimension equal to the number of ROIs. Therefore, due to this shallow decoder architecture and minimal computational overhead, the pseudo-UM version NeuroBOLT is expected to be faster than the proposed UnEBOLT model. From an efficiency perspective, the primary advantage of unified modeling lies in avoiding iterative, ROI-wise training and inference, and this benefit becomes more pronounced as the number of ROIs increases.
>
>    To avoid potential confusion or overinterpretation, we have therefore removed the corresponding time-efficiency claim from the main text and now limit our discussion to the conceptual efficiency advantage of unified whole-brain inference over iterative ROI-wise processing, without making explicit runtime comparisons across these baselines.
>
>    If we have misunderstood your question, please do let us know \- we would be happy to address it accordingly.
>
> ---
>
> >**Clearer positioning within the literature (W2)**
>
>    Thank you for your valuable suggestion. We have now included a more detailed discussion of the paper’s positioning in the Introduction. Please see the highlighted changes in the revised manuscript.
>
> ---
>
> >**Rationale for choosing the eyes-closed resting-state and further validation on the task-condition dataset (W3)**
>
>    Thank you for your valuable suggestion. In this study, we primarily focus on resting-state data, as it captures the brain’s spontaneous and intrinsic activity and exhibits greater variability in brain dynamics, which makes it inherently more challenging to decode. Training on resting-state data also encourages the model to learn intrinsic cross-modal relationships, rather than relying on task-specific effects. Moreover, resting-state paradigms are widely adopted due to their broad applicability across diverse populations, particularly in clinical settings, where task-based experiments may be impractical or unreliable for many patient groups. We have now added this discussion to the Introduction section in the revised manuscript.
>
>    To assess how well a model trained on resting-state data transfers to task-condition datasets, we now conduct comprehensive evaluations on an **auditory task-based EEG–fMRI dataset** under three settings:
>    (1) **zero-shot prediction**, where the pretrained model is frozen and directly evaluated on task-condition data;
>    (2) **fine-tuning**, where the pretrained model is further adapted using task-condition training data; and
>    (3) **training from scratch** on the task-condition dataset.
>
>    Detailed experimental settings are provided in ***Section 3.2.3 (Model Generalization Analysis)*** of the revised manuscript. As shown in ***Table 2***, the model pretrained on resting-state data achieves performance comparable to models trained within the task domain, even though the task data were collected under different experimental conditions and acquisition devices. These results highlight the potential of the proposed model for broader application to task-based datasets. A more comprehensive evaluation across diverse task paradigms is left for future work, as additional simultaneous EEG-fMRI datasets under varied task conditions become available. Due to the challenges associated with simultaneously acquiring EEG and fMRI data (including the need for specialized EEG equipment that can operate in the MRI scanner; extra setup and subject preparation time; and additional post-processing/artifact correction steps), relatively few such datasets are currently in the public domain.
>
> ---
>
> >**Subject-wise performances (W4)**
>
>    We thank the reviewer for the suggestion. Following this comment, we have added subject-level performance visualizations, where each subject is explicitly shown, to provide a more transparent view of performance stability across subjects (***see Fig.5 and Table 3 in the Appendix*** of the revised manuscript).

---

### Author Rebuttal · Authors · 2026-01-25

**Rebuttal:**

We sincerely thank the reviewers for their time and effort, as well as their constructive feedback, which have been invaluable in improving the manuscript and shaping our future work. Please find the revised manuscript in the supporting material below.

**Supporting Material:**

/attachment/a9d922d364579caa05aa275dd26f4bbf08f44148.pdf

---

### Meta-Review · Area_Chair_8Pzu · 2026-02-02

**Recommendation:** Accept (Oral)
**Confidence:** 5

**Metareview:**

All reviewers are converging towards accepting this work, with two of them recommending it be presented as an oral on the merit of its contributions and presentation.

---

### Decision · Program_Chairs · 2026-02-13

Accept (Poster)